# Phytochemical Investigation of *Myrcianthes cisplatensis*: Structural Characterization of New *p*-Coumaroyl Alkylphloroglucinols and Antimicrobial Evaluation against *Staphylococcus aureus*

**DOI:** 10.3390/plants12051046

**Published:** 2023-02-24

**Authors:** Francesca Guzzo, Elisabetta Buommino, Leslie Landrum, Rosita Russo, Francesca Lembo, Antonio Fiorentino, Brigida D’Abrosca

**Affiliations:** 1Department of Environmental Biological and Pharmaceutical Sciences and Technologies-DiSTABiF, University of Campania “Luigi Vanvitelli”, Via Vivaldi 43, 81100 Caserta, Italy; 2Department of Pharmacy, University of Naples “Federico II”, Via Domenico Montesano 49, 80131 Naples, Italy; 3School of Life Sciences, Arizona State University, Tempe, AZ 85287-4108, USA

**Keywords:** *Myrcianthes cisplatensis*, alkyphloroglucinols, 2D-NMR experiments, antimicrobial assessment, *Staphylococcus aureus*

## Abstract

Species of *Myrtaceae* Juss., the ninth largest family of flowering plants, are a valuable source of bioactive specialized metabolites. A leading position belongs to phloroglucinol derivatives, thanks to their unusual structural features and biological and pharmacological properties. *Myrcianthes cisplatensis* (Cambess.) O. Berg, a common tree on the banks of rivers and streams of Uruguay, southern Brazil, and northern Argentina, with aromatic leaves, is known as a diuretic, febrifuge, tonic, and good remedy for lung and bronchial diseases. Despite knowledge about traditional use, few data on its phytochemical properties have been reported in the literature. The methanol extract of *M. cisplatensis*, grown in Arizona, USA, was first partitioned between dichloromethane and water and then with ethyl acetate. The enriched fractions were evaluated using a broth microdilution assay against *Staphylococcus aureus* ATCC 29213 and 43300 (methicillin-resistant *S. aureus* (MRSA)). The potential antimicrobial activity seemed to increase in the dichloromethane extract, with a MIC value of 16 µg/mL against both strains. Following a bio-guided approach, chromatographic techniques allowed for isolating three coumarin derivatives, namely endoperoxide G3, catechin, and quercitrin, and four new *p*-coumaroyl alkylphloroglucinol glucosides, named *p*-coumaroylmyrciacommulone A-D. Their structures were characterized through spectroscopic techniques: 2D-NMR experiments (HSQC, HMBC, and HSQC-TOCSY) and spectrometric analyses (HR-MS). The antimicrobial assessment of pure compounds against *S. aureus* ATCC 29213 and ATCC 43300 demonstrated the best activity for *p*-coumaroylmyrciacommulone C and D with the growth inhibition of 50% at 32 µg/mL against both strains of *S. aureus*.

## 1. Introduction

*Myrcianthes cisplatensis* (Cambess.) O. Berg is a native plant to northern Argentina, southern Brazil, and Uruguay, found at elevations of sea level to ca. 1000 m; it frequently grows along streams. It possibly grows in Paraguay, but no specimen records have been found. The genus *Myrcianthes*, a group of about 40 species, widely distributed in South America, is characterized by having an embryo with thick unfused plano-convex cotyledons and uniflorous or dichasial inflorescences [1]. Its fruits are edible berries eaten by birds and mammals and are used for the preparation of marmalades [2,3]. The genus belongs to the large family of *Myrtaceae*, which comprises ca. 5700 species with concentrations of diversity in South America and Australasia (WCSP) [4].

*M. cisplatensis* is reported to be used in traditional medicine as a stimulant, diuretic, and diaphoretic [5]. Despite its documented uses by indigenous people as food and medicine [6], *M. cisplatensis* is underexplored from a phytochemical point of view. Previously, essential oils have been investigated with 1,8-cineole and limonene reported as major components of the leaf oil [7,8]. Additionally, conglomerone, a phloroglucinol derivative, has been reported as a constituent of its leaves [9]. This compound exhibited antimicrobial activity against *Staphylococcus aureus* with a MIC of 62.5 µg/mL for the sensitive strain (6538p) and MIC of s250 µg/mL for the multiresistant strains (43300 and 700699). Recent decades, in fact, have witnessed a remarkable increase in studies reporting antimicrobial compounds against pathological microorganisms isolated from *Myrtaceae* species [10]. To name just a few phloroglucinols, Myrtocummulone A from *Myrtus communis* [10], Callistrilone A from *Callistemon lanceolatus* [11], Rhodomyrtone A from *Rhodormyrtus tomentosa* [12], eugenials C and D from *Eugenia umbrellifora* [13], and, recently, four new alkylphloroglucinol glucosides isolated from *M. communis* have been found to be active against the *S. aureus* MRSA strain [14]. Fortunately, panels of new alkylphloroglucinols with antimicrobial activity are being discovered, which is very interesting due to the emergence of multidrug-resistant strains such as MRSA [15]. Antimicrobial resistance is one of the greatest challenges for scientists since it still represents a serious global public health threat, killing at least 1.27 million people worldwide. The spread of new drug-resistant strains of bacteria has diminished previous efforts in the discovery of new antimicrobials. Thus, it is of great importance to pursue research on natural compounds, derived from plants, especially Myrtaceae, with a proven record of antimicrobial properties. Here, we investigated for the first time the antibacterial potential of the methanolic extract of *M. cisplatensis* leaves against two *S. aureus* ATCC strains (29213 and 43300). Following a bio-guided investigation, four new *p*-coumaroyl alkylphloroglucinol glucosides, named *p*-coumaroylmyrciacommulone A-D, were isolated and characterized, via 2D-NMR, for the first time. The pure compounds *p*-coumaroylmyrciacommulone C and D were strongly active against both *S. aureus* strains. This paper is the fourth report about the presence of alkylphloroglucinol glucosides in plants belonging to the *Myrtaceae* family.

## 2. Results and Discussion

### 2.1. Antibacterial Assessment of Methanolic Extract and First Purification Step

The crude extract of *M. cisplatensis* (MYR) was tested for its antimicrobial potential against *S. aureus* ATCC 29213 and *S. aureus* ATCC 43300 (methicillin-resistant strain (MRSA)) using a broth microdilution assay. The extract showed promising activity on both strains, reporting the same inhibitory effects. MYR strongly reduced the growth of both strains (about 83%) at 64 μg/mL; a good antimicrobial activity was kept until the concentration of 16 μg/mL (Figure 1). The positive control, vancomycin, strongly inhibited the growth at 2 μg/mL (88.8% ± 2.1), whereas oxacillin was not able to inhibit the growth of the MRSA. The negative control methanol did not affect the cell growth at any of the tested dilutions.

Based on these promising results, MYR was further purified through liquid–liquid separation (Figure 2) first with dichloromethane (MYR_A) and then using ethyl acetate (MYR_B). The medium-polar fraction (MYR_A) and the more-polar one (MYR_B) were characterized using different metabolite contents, confirmed by their ^1^H-NMR profiling. In particular, MYR_A showed prevailing overlapped signals in the aliphatic region (1.6–0.2 ppm). Initially, the presence of fatty acids (Figure 2A) made it difficult to understand what kind of compounds were present in MYR_A. On the other hand, the ^1^H-NMR spectrum of MYR_B was characterized by resonances typical of catechin (Figure 2B). In addition, in the aromatic region (7.5–6.0 ppm), the presence of ^1^ the H-NMR spectrum of two additional meta-coupled doublets at δ_H_ 6.36 and δ_H_ 6.20, as well as the overlapped signals in the region between 5.0 and 3.3 ppm suggested the presence of flavonol glycosides in MYR_B.

The two fractions were tested for their antimicrobial properties. MYR_A showed the best activity on both *S. aureus* strains, reducing the growth by 87% at the lowest concentration of 16 μg/mL (Figure 2C), thus leading to more promising results with respect to the parental extract MYR. By contrast, MYR_B was strongly active at 128 μg/mL, reducing the growth of both *Staphylococci* strains by 84% (±2) but gradually decreased in activity at lower concentrations, being lowest at 16 μg/mL with the growth inhibition of 43% (± 1) and 30% (±1) against *S. aureus* 29213 and MRSA, respectively (Figure 2C).

### 2.2. Purification of MYR_A and NMR Characterization of p-Coumaroyl Alkylphloroglucinols

A metabolic complexity of MYR_A required purification in order to perform the complete structural characterization of this medium-polar fraction. The HPLC, RP-18 led to the isolation of four *p*-coumaroyl alkylphloroglucinols (**1–4**, Figure 3), isolated for the first time in addition to endoperoxide G3 (**5**) and three coumarin derivatives (**6–8**).

Compound **1,** named *p*-coumaroylmyrciacommulone A, was described for the first time. It showed a molecular formula C_29_H_38_O_11_ in accordance with the pseudo-molecular [M + K]^+^ and [M + Na]^+^ peaks in the ESI Q-TOF HRMS at *m/z* 610.21 and *m/z* 585.23, respectively, and with the ^13^C-NMR data. In fact, this spectrum showed 29 carbons identified, on the basis of an HSQC experiment (Table 1), as 6 methyls, 1 methylene, 13 methines, and 9 quaternary carbons, including 2 ketones (δ_C_ 211.2 and 216.7) and 1 ester (δ_C_ 169.1). The ^1^H-NMR spectrum (Table 1) was dominated by the signals of the *p*-coumaroyl moiety suggested by the presence of an AA’BB’ system, as two aromatic doublets at δ_H_ 6.81 (H-2″/H-6″, δ_C_ 116.9) and δ_H_ 7.46 (H-3″/H-5″, δ_C_ 131.0), as well as two olefinic doublets at δ_H_ 7.63 (H-7″δ_C_ 146.8) and δ_H_ 6.35 (H-8″δ_C_ 114.6). The ion at *m/z* 147.04, present in the ESI Q-TOF MS/MS spectrum, confirmed the presence of this moiety. The coupling constant value of 15.9 Hz for olefinic protons was in good agreement with an *E* configuration for the double bond of *p*-coumaroyl moiety.

An olefinic proton at δ_H_ 5.10 (δ_C_ 118.7), an oxymethine proton at δ_H_ 3.95 (δ_C_ 89.1), and four singlets in the upfield region of the ^1^H-NMR spectrum, at δ_H_ 1.23 (δ_C_ 25.3), 1.31 (δ_C_ 26.1), 1.31(δ_C_ 28.7), and 1.26 (δ_C_ 27.2), were also evident. Two singlets at δ_H_ 1.77 (δ_C_ 20.1) and 1.68 (δ_C_ 27.3) suggested the presence of two methyls bonded to sp^2^ carbon. Finally, the presence of a doublet at δ_H_ 4.60 (δ_C_ 106.4) together with other overlapped signals in the 3.00–4.60 ppm region were in good agreement with the presence of a sugar moiety.

The sequence of the glycosidic moiety was established based on the two-bond hetero-correlations evident in the H2BC spectra and also supported by the HSQC (Figure 4A) data, suggesting the presence of a glucose moiety. The β configuration of C-1′ carbon was determined on the basis of the coupling constant value (*J* = 8.1 Hz) of the anomeric proton. The downfield-shifted values of methylene carbon (C-6′), as well as of the H-6′ doublet of the doublets of glucose (δ_H_ 4.51/4.37 and δ_C_ 64.2) and the correlations of these latter with carboxyl carbon at δ_C_ 169.1, highlighted in the HMBC experiment (Figure 4B), allowed the localization of *p*-coumaroyl moiety at the C-6′ carbon of the sugar. 

In the HMBC experiment (Figure 4B), the anomeric proton at δ_H_ 4.60 showed hetero-correlations with C-10 (δ_C_ 89.1) methine carbon, whose proton at δ_H_ 3.95 showed hetero-correlations with both ketone carbons C-6 (δ_C_ 211.2) and C-8 (δ_C_ 216.7), as well as with C-7 and C-9 quaternary carbons (δ_C_ 48.7 and δ_C_ 57.3, respectively) and with C-5 quaternary carbinol (δ_C_ 83.4), confirming the presence of two hydroxyls on the cyclohexadione skeleton [16], while the hetero-correlation with C-4 methine at δ_C_ 118.7 allowed us to localize the isobutylene moiety at the C-5 of cyclohexadione (Figure 3).

The *cis* configuration of vicinal hydroxyl groups was determined based on the observed NOE, in the NOESY experiment, among the H-4 vinyl proton and both the H-10 proton and H-14 (Appendix A). Moreover, the NOE between the H-10 proton and H-14 methyl allowed us to deduce a β orientation for C-10. All these spectroscopic data confirmed the hypothesized structure for *p*-coumaroylmyrciacommulone A (**1**)**.** Compound **2** was identified as *p*-coumaroylmyrciacommulone B, namely a *Z*-isomer at the double bond of *p*-coumaroyl moiety of compound **1**. It showed a molecular formula C_29_H_38_O_11_, in good agreement with the quasi-molecular ion [M-Na]^+^ at *m/z* 585.23 and the fragment of *p*-coumaroyl moiety ion at *m/z* 147.04. The NMR spectra of compounds **2** and **1** showed the same ^1^H spin system and ^13^C multiplicities but showed significant differences for the olefinic protons of the *p*-coumaroyl moiety. In fact, the upfield-shifted values of protons H-8″ at δ_H_ 5.78 (δ_C_ 116.4) and H-7″ δ_H_ 6.88 (δ_C_ 145.7) were evident in the ^1^H-NMR spectrum. Furthermore, the coupling constant value (*J* = 12.3 Hz) for olefinic protons H-8″ and H-7″ was in accordance with a *cis-* geometry for a double bond. These latter protons showed hetero-correlations with carboxyl carbon C-9″ a δ_C_ 168.2, which were, in turn, hetero-correlated with methylene protons H-6′ (δ_H_ 4.49/4.32) of the glucose unit. The hetero-correlation between the anomeric proton H-1′ and carbon C-10 (δ_H_ 3.91) suggested the presence of a dihydroxy cyclohexadione skeleton, whose C-10 was linked to a glucose-6″ *p*-coumaroyl unit (Table 1, Figure 5B).

The *cis* configuration of vicinal dihydroxyl was determined based on the observation of the NOE between the oxymethine proton H-10 and vinylic proton H-4, as well as between the latter and methyl protons H-12 and H-14. Finally, the NOE was also evident between the oxymethine proton H-10 and the anomeric proton (Figure 5A).

Compound **3,** named *p*-coumaroylmyrciacommulone C, was described for the first time. It showed a molecular formula of C_29_H_40_O_11_, as suggested by quasi-molecular ion [M-Na]^+^ at *m/z* 587.23. The NMR data revealed significant differences related to an isobutylene unit. In fact, the ^1^H-NMR experiment, along with typical resonances of *p*-coumaroyl moiety (Table 2), was characterized by the presence of a doublet of the doublets of a methylene proton at δ_H_ 1.61 (δ_C_ 25.7), a multiplet of a methine proton at δ_H_ 1.90 (δ_C_ 45.3), and two doublets of two methyl groups at δ_H_ 0.88 (δ_C_ 25.5) and δ_H_ 0.93 (δ_C_ 25.6).

The cross-peaks observed in the COSY experiment indicated the correlations between H-1 and H-2 methyl protons and H-3 methine, as well as between the latter and H-4 methine protons. In the HMBC experiment (Appendix A, Table 2), the presence of isobutyl moiety was confirmed by the hetero-correlations shown between the methine proton at δ_H_ 1.61 (H-3) and carbons at δ_C_ 25.5 (C-1), 25.6 (C-2), 45.3 (C-4), and 85.9 (C-5). Moreover, the heterocorrelations between methylene protons at δ_H_ 1.90 (H-4) and carbons at δ_C_ 25.5 (C-1), 25.6 (C-2), 25.7 (C-3), 85.9 (C-5), and 212.5 (C-6) suggested the presence of an alkylation site at the C-5 carbon of the cyclohexadione skeleton. The latter was highlighted by typical hetero-correlations among methyl protons at δ_H_ 1.17 (H-11), 1.22 (H-12), 1.36 (H-13), and 1.23 (H-14), and ketone carbons at δ_C_ 212.5 (C-6), 214.5 (C-8), and with quaternary carbons at δ_C_ 50.4 (C-7) and δ_C_ 51.2 (C-9). Finally, in the same experiment, the correlation between the anomeric proton at δ_H_ 4.48 (H-1′) and carbon at δ_C_ 87.3(C-10) allowed for the localization of 6′-*O*-*p*-coumaroyl glucose moiety at C-10. These spectra were in good accordance with the hypothesized structure for *p*-coumaroylmyrciacommulone C (**3**).

Compound **4** showed similar ^1^H spin systems and ^13^C multiplicities (Table 2) to those of compound **3**. In particular, the coupling constant value for the olefinic proton at δ_H_ 5.78 (*J* = 12.8) defined compound **4** as a *cis* isomer of compound **3**. All these spectroscopic data (Table 2) confirmed the hypothesized structure for *p*-coumaroylmyrciacommulone D (**4**), described for the first time in the present paper.

The NMR data of compound **5** (Figure 3) were in good agreement with those reported in the literature for endoperoxide G3 [17]. Finally, compounds **6–8** were identified as 5,7-dihydroxy-4-methylcoumarin (**6**), 5,7-dihydroxy-4,6-dimethylcoumarin (**7**), and 5,7-dihydroxy-4,8-dimethylcoumarin (**8**) [18].

### 2.3. The 2D-NMR Characterization of MYR_B

A 2D-NMR investigation was helpful in the characterization of the flavonoid compound of the more polar fraction MYR_B. In particular, NMR data allowed us to confirm the presence of catechin (**9**) [19]. Remaining to define were the second main flavonoidic compound. The 2D-NMR data suggested the presence of quercetin moiety as aglycon linked to rhamnose unit thanks to the long-range heterocorrelations (Appendix A) between the anomeric proton at δH 5.36 (H-1″) and a quaternary carbon compound at δC 134.0 (C-3). All these data allowed for the characterization of this compound as a 3-*O*-rhamnosyl quercetin (**10**).

Pure **9** and **10**, obtained via MYR_B, were tested for their antimicrobial activity against *S. aureus* 25923 and MRSA.

### 2.4. Antimicrobial Assessment of Pure Compounds

Pure compounds **1–4**, **6**, and **9–10** were investigated for their antimicrobial properties against *S. aureus* ATCC 29213 and ATCC 43300 (Figure 6).

Catechin (**9**), quercitrin (**10**), and 5,7-dihydroxy-4-methylcoumarin (**6**) were inactive (data not shown). By contrast, *p*-coumaroyl alkylphloroglucinol (**2–4**) inhibited growth in the concentration range of 32 µg/mL to 128 µg/mL (Figure 6). In particular, the best antimicrobial activity was shown by compounds **3** and **4**, evaluated in the mixture, with the growth inhibition of 50% at 32 µg/mL against both *S. aureus* strains. The antimicrobial activity of compound **2** was also of great interest, reducing growth at 128 µg/mL (53% ± 4 and 55% ± 4) and 64 µg/mL (55% ± 4, and 51% ± 4) against ATCC 29213 and MRSA, respectively. By contrast, compound **1** reported weak activity only on *S. aureus* ATCC 29213 at 32 µg/mL. This result might reflect the chemical structures of the tested compounds. It was interesting to notice that the E/Z isomers **1** and **2** affected *S. aureus* growth differently. In fact, the best activity was reported by compound **2**, which presented a *cis*- configuration at the double bond of *p*-coumaroyl moiety. The literature data confirm these results. In fact, a double-bond geometry influences the antimicrobial properties of different etherolenic acid isomers against phytopathogenic bacteria with the Z isomer more active than the E isomer [20]. In addition, fatty acids with *cis* double bonds have more pronounced antibacterial properties than fatty acids with *trans* double bonds [21,22]. So, the potential of compound **2** might depend on this geometric isomerism. Furthermore, the antimicrobial results of compounds **2**, **3**, and **4** were in good agreement with the antibacterial activity reported for the related galloylated alkylphloroglucinol glucosides [14,16], whose moderate effect was reached in the concentration range of 64 µg/mL to 256 µg/mL.

## 3. Materials and Methods

### 3.1. Plant Material

Leaves of *M. cisplatensis* were collected at the flowering stage (May 2022) in Arizona, USA (Maricopa Country, Tempe, 9308 S. Margo Dr, 370 m 1200 ft) and identified by prof. Leslie Landrum. A voucher specimen (L.R. Landrum 11697) was deposited at the ASU Vascular Plant Herbarium. Seeds of the voucher plant were collected near Colonia, Uruguay. *M. cisplatensis* is not protected by local or international regulations; therefore, no specific permission was required for its collection. The leaves were dried in a ventilated stove at 40 °C to constant weight, powdered with liquid nitrogen, and stored at −20 °C until the next analysis.

### 3.2. General Chromatographic Procedures

Analytical TLC was performed on Merck Kieselgel 60 F254 or RP-8 F254 plates of 0.2 mm layer thickness. The plate was visualized using UV light or by spraying with H_2_SO_4_/AcOH/H_2_O (1:20:4), followed by heating at 120 °C for about 1 min. The plates were then heated for 5 min at 120 °C. Preparative TLC was performed on Merck Kieselgel 60 F254 plates, of 0.5 or 1 mm film thickness. Column chromatography (CC) was performed on: Sephadex LH-20 (Sigma, Merck Kieselgel 60 (70–240 µm), Merck Kieselgel 60 (40–63 µm), and Baker (Deventer, Netherlands) RP-18. The preparative HPLC apparatus consisted of Knauer Smartline 31/40 module equipped with a Knauer Smartline 1000 pump, UV detector, and PC Crom-Gate software (Knauer, Berlin, Germany). Preparative HPLC was performed using Kromasil RP-18 (10 lm, 250 10.0 mm i.d., Phenomenex, Torrance, CA, USA).

### 3.3. NMR Experiments

NMR spectra were recorded at 25 °C on 300.03 MHz for ^1^H and 75.45 MHz for ^13^C on a Bruker AVANCE II 300 MHz NMR Spectrometer Fourier transform in CD_3_OD or CDCl_3_ (Bruker, Billerica, MA, USA). Chemical shifts are reported in δ (ppm) and referenced to the residual solvent signal; J (coupling constant) is reported in Hz.^1^H-NMR spectra were acquired over a spectral window from 14 to −2 ppm, with 1.0 s relaxation delay, 1.70 s acquisition time (AQ), and 90° pulse width = 13.8 µs. The initial matrix was zero-filled to 64 K. The ^13^CNMR spectra were recorded in the ^1^H broadband decoupling mode, over a spectral window from 235 to −15 ppm, 1.5 s relaxation delay, 90° pulse width = 9.50 µs, and AQ = 0.9 s. The number of scans for both ^1^H and ^13^C-NMR experiments was chosen, depending on the concentration of the samples. With regard to the homonuclear and heteronuclear 2D-NMR experiments, the data points, number of scans, and increments were adjusted according to the sample concentrations. Correlation spectroscopy (COSY) spectra were recorded with a gradient-enhanced sequence at spectral widths of 3000 Hz in both f2 and f1 domains; the relaxation delays were 1.0 s. Nuclear Overhauser effect spectroscopy (NOESY) experiments were performed in the phase-sensitive mode. The mixing time was 500 ms, and the spectral width was 3000 Hz. For all the homonuclear experiments, an initial matrix of 512 × 512 data points was zero-filled to give a final matrix of 1 k × 1 k points. Proton-detected heteronuclear correlations were also measured. Heteronuclear single-quantum coherence (HSQC) experiments (optimized for ^1^J_(H,C)_ = 140 Hz) were performed in the phase-sensitive mode with field gradient. The spectral width was 12,000 Hz in f1 (^13^C) and 3000 Hz in f2 (^1^H) and had a 1.0 s relaxation delay; a matrix of 1 k × 1 k data points was zero-filled to give a final matrix of 2 k × 2 k points. Heteronuclear 2-bond correlation (H2BC) spectra were obtained with T = 30.0 ms and a relaxation delay of 1.0 s; a third-order low-pass filter was set for 130 < ^1^J_(C,H)_ < 165 Hz. A heteronuclear multiple-bond coherence (HMBC) experiment (optimized for ^1^J_(H,C)_ = 8 Hz) was performed in the absolute value mode with field gradient; typically, ^1^H–^13^C gHMBC were acquired with a spectral width of 18,000 Hz in f1 (^13^C) and 3000 Hz in f2 (^1^H) and 1.0 s of relaxation delay; the matrix of 1 k × 1 k data points was zero-filled to give a final matrix of 4 k × 4 k points. Heteronuclear single-quantum coherence–total correlation spectroscopy (HSQC-TOCSY) experiments were optimized for ^n^J_(H, C)_ = 8 Hz, with a mixing time of 90 ms.

### 3.4. High-Resolution MS and MS/MS Analysis

For accurate mass measurements, the purified compounds were analyzed using a quadrupole time-of-flight (Q-TOF) SYNAPT G2Si High-Definition Mass Spectrometer (Waters, Manchester, UK), equipped with an electrospray ionization (ESI) lock-spray source and a low-flow probe [23]. All the analyses were carried out in a positive-continuum ion mode with the spray voltage set at 3 kV, by using a sampling cone voltage of 40 V and a source offset of 80 V. The source temperature was kept at 120 °C, and nitrogen was used as drying gas (flow rate of 300 L/h).

MS and MS/MS spectra (*m*/*z* range 50–1500) were acquired in the resolution mode (20,000 resolution full width at half maximum at *m*/*z* 400) with a fixed acquisition time of 60 s. Argon gas was used for collision-induced dissociation (CID). The time-of-flight analyzer of the mass spectrometer was externally calibrated with Glu-fibrinopeptide (100 fmol/μL in CH_3_CN:H_2_O 50:50 (*v*/*v*) + 1% formic acid) from *m*/*z* 50–1600.

Accurate mass data were collected by directly infusing the samples (1.5 pmol/μL in CH_3_OH) into the system at a flow rate of 5 μL/min. The acquisition and processing of the data were performed with the MassLynx 4.1 software (Waters, Manchester, UK). Compound (**1**): HRMS (ESI) calcd for C_29_H_38_O_11_K [M + K]^+^: 601.7, found 601.21, calcd for C_29_H_38_O_11_Na [M + Na]^+^: 585.6 found 585.23. Compound (**2**): HRMS (ESI) calcd for C_29_H_38_O_11_Na [M + Na]^+^: 585.6 found 585.23. Compound (**3**): HRMS (ESI) calcd for C_29_H_40_O_11_Na [M + Na]^+^: 587.62 found 587.25.

### 3.5. Compound Purification

Dried leaf material (100 g) was powdered and extracted via ultrasound-assisted extraction (Branson 3800 MH) for 40 min each and in methanol (2 cycles × 3.0 L). After centrifugation at 4800× *g* rpm (Beckman, GS-15R) for 10 min at 4 °C, the solution was dried in vacuum, giving 12.4 g of the crude extract. The methanolic crude extract was dissolved in H_2_O and extracted with dichloromethane in a separatory funnel. The water phase was successively extracted with ethyl acetate. The dichloromethane fraction MYR_A (2.4 g) was chromatographed on Sephadex LH-20, eluting with hexane/CHCl_3_/CH_3_OH (1:1:1), furnishing two fractions: MYR_A_1_ and MYR_A_2_. The fraction MYR_A_1_ was chromatographed via RP-18 HPLC eluted with H_2_O/CH_3_OH/CH_3_CN (5:2:3), giving the pure compounds **1** (6.0 mg), **2** (3.9 mg), and a mixture of **3** and **4** (3.1 mg). The fraction MYR_A_2_ was chromatographed on RP-18 CC, eluted with H_2_O/CH_3_OH/CH_3_CN solutions at decreasing polarity, which furnished pure compounds **5** (4.5 mg), **6** (1.0. mg), **7** (5.9 mg), and **8** (2.2 mg). An aliquot (30 mg) of MYR_B was chromatographed on TLC preparative (0.50 mm) with CH_3_OH/CHCl_3_ (1:4) as the eluent, furnishing compounds **9** (6.0 mg) and **10** (4.1 mg).

### 3.6. Antimicrobial Test

#### 3.6.1. Microorganism and Growth Conditions

*S. aureus* ATCC 29213 and *S. aureus* ATCC 43300 were obtained from the American Type Culture Collection (Rockville, MD, USA). All the strains were cultured in tryptic soya broth (TSB, OXOID) under aerobic conditions at 37 °C for 24 h on an orbital shaker at 200 rpm. Each tested compound was dissolved in methanol (1 mg/200 μL) to give a stock solution and diluted in Mueller–Hinton (MH).

#### 3.6.2. Susceptibility Assay on Bacteria Planktonic Cells

Minimal inhibitory concentrations (MICs) of the tested compounds were determined in a specific medium using the broth microdilution assay, according to the European Committee on Antimicrobial Susceptibility Testing (EUCAST version 7.1, 7 June 2017), as previously reported [14]. Briefly, 100 μL of bacterial culture was diluted in MH broth at a final cell concentration of 1 × 10^6^ CFU/mL and added to each well of a 96-well plate containing 100 μL of MH with serially two-fold diluted compounds ranging from 128 to 16 µg/mL. Each MIC value indicates the lowest concentration of the compound that causes a total inhibition of bacterial growth after 24 h of incubation at 37 °C. Negative control wells were set to contain bacteria in the Mueller–Hinton broth plus the amount of vehicle (methanol) used to dilute each compound. The positive controls included vancomycin (2 µg/mL) and oxacillin (2 µg/mL).

#### 3.6.3. Statistical Analysis

All the results of antimicrobial activity are expressed as the means ± standard deviation (S.D.). The results were analyzed using one-way analysis of variance (ANOVA) followed by Tukey’s post hoc comparison tests to verify the differences between compounds and concentrations (*p* < 0.05).

## 4. Conclusions

The bio-guided phytochemical method is still the basic commonly accepted procedure for characterizing new natural products that are pharmacologically active. Following this approach, natural compounds with potent antimicrobial activity were isolated from the plants belonging to the *Myrtaceae* family. In particular, alkylphloroglucinols are responsible for the documented antimicrobial properties of the family. In this context, here, we reported the results of a phytochemical investigation of the methanol leaf extract of *M. cisplatensis* growing in Arizona. This led, for the first time, to the isolation and characterization of four *p*-coumaroylmyrciacommulones A-D. Unlike the wide distribution of alkylphloroglucinol in plants belonging to the *Myrtaceae* family, alkylphloroglucinol glucosides are reported as constituents of only few *Myrtaceae* plants. From both phytochemical and biological points of view, these results are of great interest since they pave the way for future studies. In particular, with docking molecular analysis, we plan to pursue an investigation of more active alkylphloroglucinols in order to provide some information about their interaction mechanism.

## Figures and Tables

**Figure 1 plants-12-01046-f001:**
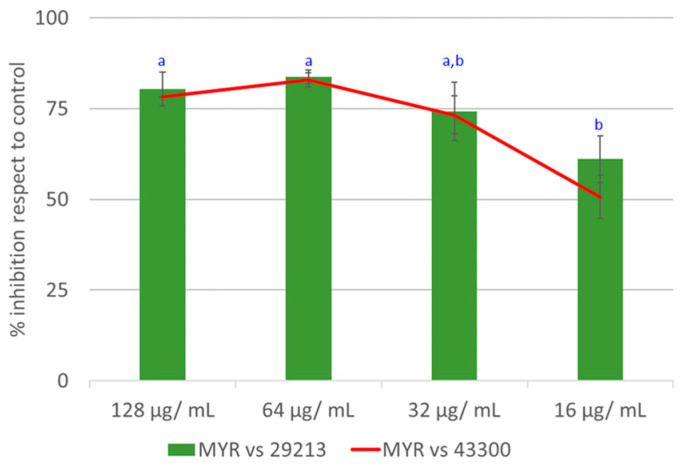
Antimicrobial assay of methanolic extract of *M. cisplatensis* against *S. aureus* ATCC 29213 and ATCC 43300 strains. One-way ANOVA with Tukey multiple comparisons: different letters indicate significant differences among groups for each strains (^a^ *p* < 0.01 and ^b^ *p* < 0.05).

**Figure 2 plants-12-01046-f002:**
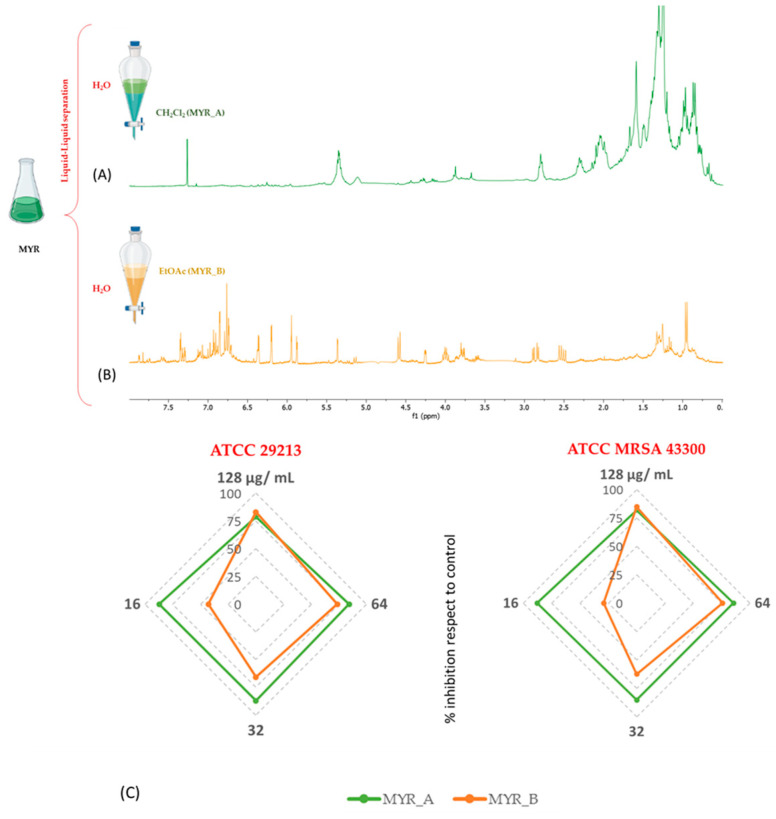
Liquid–liquid separation of methanolic extract and ^1^H-NMR profiling of organic fractions (**A**,**B**). Antimicrobial screening of MYR_A and MYR_B (**C**).

**Figure 3 plants-12-01046-f003:**
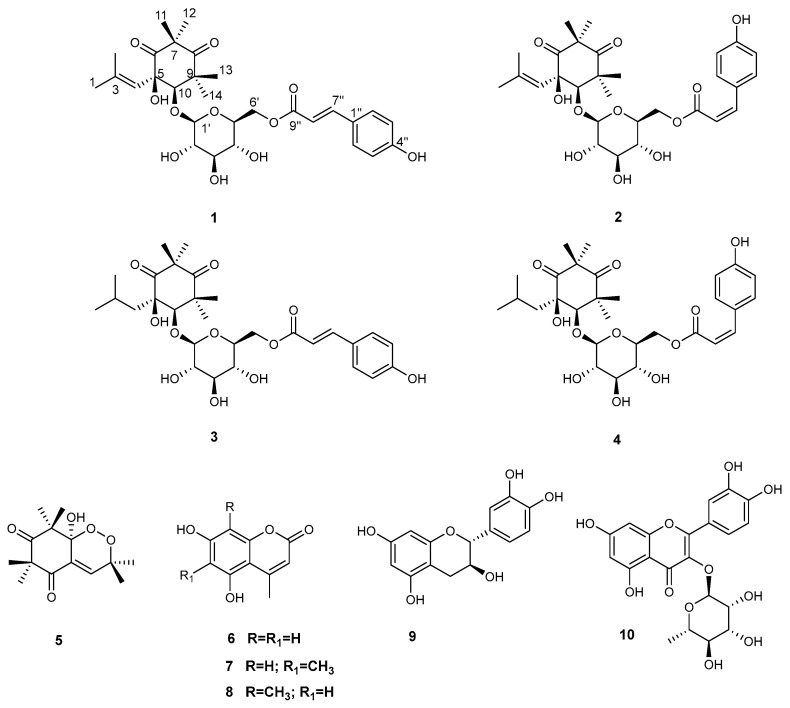
Chemical structure of compounds **1–10**, isolated from leaves of *M. cisplatensis*.

**Figure 4 plants-12-01046-f004:**
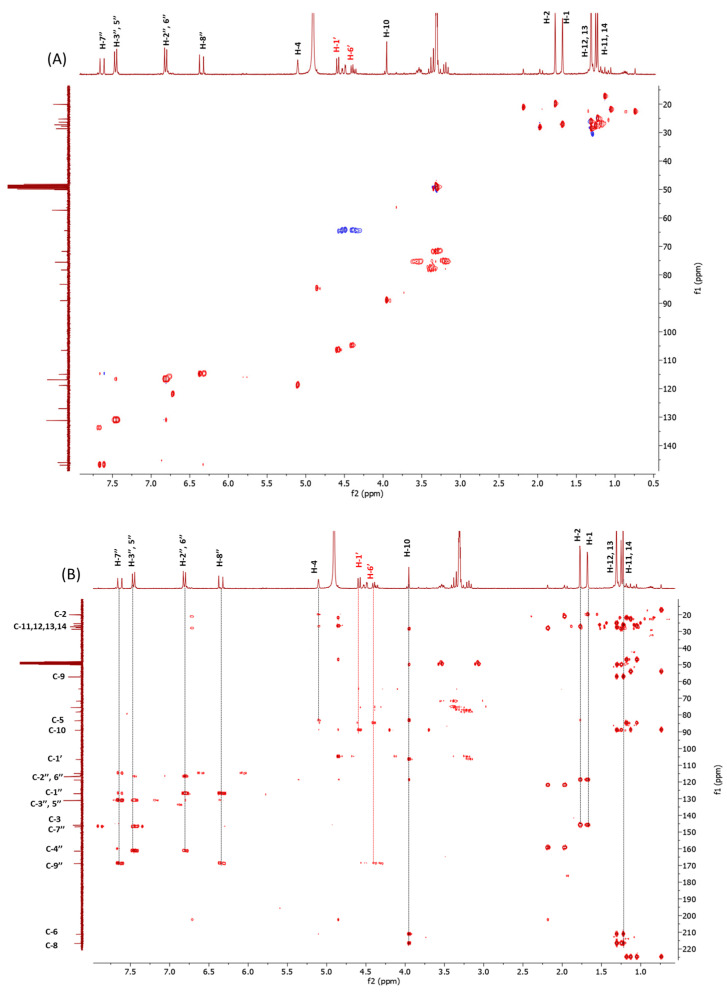
HSQC (**A**) and HMBC (**B**) experiments of compound **1**.

**Figure 5 plants-12-01046-f005:**
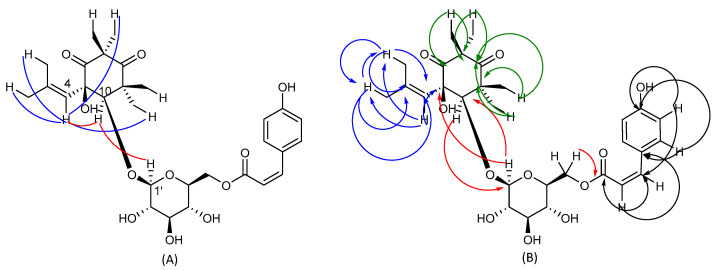
Selected NOESY correlation (**A**) and H-C long-range correlations (**B**) of compound **2** evidenced in HMBC. Blue, red, green and black arrows are referred to correlelations of isobutylene, cyclohexadione, *p*-coumaroyl and glucose moiety, respectively.

**Figure 6 plants-12-01046-f006:**
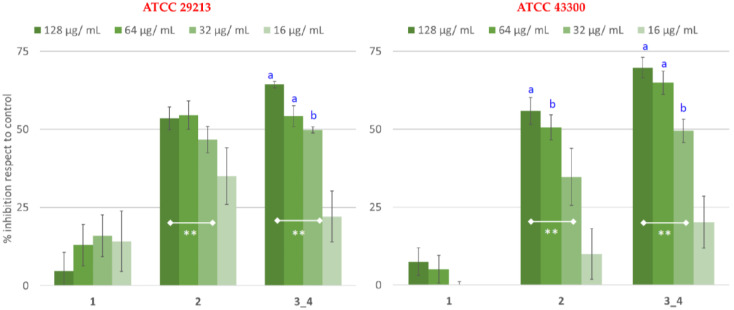
Antimicrobial screening of pure compounds isolated from bioactive fractions of *M. cisplathensis.* One-way ANOVA with Tukey multiple comparisons. ^a^
*p* < 0.01 and ^b^
*p*< 0.05 indicate significant results for compounds **2** and **3**_**4** at 128, 64, and 32 μg/mL compared with 16 μg/mL; ** *p* < 0.01 indicates significant results among compounds at certain concentrations.

**Table 1 plants-12-01046-t001:** NMR data of compounds **1–2** in CD_3_OD ^1^.

	*p*-Coumaroylmyrciacommulone A (1)				*p*-Coumaroylmyrciacommulone B (2)			
Position	δc (Type)	δ_H_ (J in Hz)	HMBC (H→C)	NOESY	δc (Type)	δ_H_ (J in Hz)	HMBC (H→C)	NOESY
**1**	27.3 (CH_3_)	1.68 s	3,4		27.6 (CH_3_)	1.70 s	2,3,4	
**2**	20.1 (CH_3_)	1.77 s	3,4		20.2 (CH_3_)	1.78 s	1,3,4	
**3**	145.9 (C)	-	-		145.5 (C)	-	-	
**4**	118.7 (CH)	5.10 s	1,2,5	1,12,10	118.6 (CH)	5.10 s	1,2,5	1,10,12,14
**5**	83.4 (C)	-	-		81.6 (C)	-	-	
**6**	211.2 (C)	-	-		214.8 (C)	-	-	
**7**	48.7 (C)	-	-		48.9 (C)	-	-	
**8**	216.7 (C)	-	-		209.6 (C)	-	-	
**9**	57.3 (C)	-	-		56.8 (C)	-	-	
**10**	89.1 (CH)	3.95 s	4,5,6,7,8, 13,1′	2,4,12,1′	89.7 (CH)	3.91 s	4,5,6,7,8, 11,1′	2,4,12,1′
**11**	27.2 (CH_3_)	1.26 s	5,8,7,10,12		27.7 (CH_3_)	1.20 s	7,8,10,12	
**12**	28.7 (CH_3_)	1.31 s	6,7,10,11		28.6 (CH_3_)	1.28 s	6,7,10,11	
**13**	26.1 (CH_3_)	1.31 s	8,9,10,14		26.8 (CH_3_)	1.31 s	6,8,9,14	
**14**	25.3 (CH_3_)	1.23 s	6,8,9,10,13		26.1 (CH_3_)	1.23 s	6,8,9,11	
**1′**	106.4 (CH)	4.60 d (8.1)	5,10	10	106.8 (CH)	4.55 d (7.8)	5,10	10
**2′**	75.2 (CH)	3.19 dd (8.4, 8.1)			75.9 (CH)	3.18 dd (7.8, 9.0)		
**3′**	78.3 (CH)	3.37 ov			78.2 (CH)	3.37 ov		
**4′**	71.8 (CH)	3.30 ov			72.0 (CH)	3.25 ov		
**5′**	75.5 (CH)	3.54 ov			75.4 (CH)	3.49 ov		
**6′**	64.2 (CH_2_)	4.51 dd (3.0, 12.3); 4.37 dd (6.9, 12.3)	9″		64.3 (CH_2_)	4.49 dd (2.1, 12.0); 4.32 dd (7.2, 12.0)	9″	
**1″**	127.0 (C)	-			127.3 (C)	-		
**2″**	116.9 (CH)	6.81 d (8.4)	1″,4″	3″,5″	116.1 (CH)	6.76 d (8.7)	1″,4″	3″,5″
**3″**	131.0 (CH)	7.46 d (8.4)	2″,4″, 6″,7″,9″	2″,6″,8″	134.1 (CH)	7.67 d (8.7)	2″,4″,6″, 7″,9″	2″,6″,7″
**4″**	162.6 (C)	-			160.5 (C)	-		
**5″**	131.0 (CH)	7.46 d (8.4)	2″, 4″, 6″,7″,9″	2″,6″,8″	134.1 (CH)	7.67 d (8.7)	2″,4″,6″, 7″,9″	2″,6″,7″
**6″**	116.9 (CH)	6.81 d (8.4)	1″,4″	3″,5″	116.1 (CH)	6.76 d (8.7)	1″,4″	3″,5″
**7″**	146.8 (CH)	7.63 d (15.9)	1″,3″,8″,9″	3″,5″,8″	145.7 (CH)	6.88 d (12.3)	8″,9″	3″,5″,8″
**8″**	114.6 (CH)	6.35 d (15.9)	1″,9″	3″,5″	116.4 (CH)	5.78 d (12.3)	1″,7″,9″	7″
**9″**	169.1 (C)	-			168.2 (C)	-		

^1^ d = doublet, dd = doublet of doublets; m = multiplet; ov = overlapped; s = singlet; t = triplet; *J* values (Hz) are reported in brackets.

**Table 2 plants-12-01046-t002:** NMR data of compounds **3** and **4** in CD_3_OD ^1^.

	*p*-Coumaroylmyrciacommulone C (3)			*p*-Coumaroylmyrciacommulone D (4)		
Position	δc (Type)	δ_H_ (J in Hz)	HMBC (H→C)	δc (Type)	δ_H_ (J in Hz)	HMBC (H→C)
**1**	25.5 (CH_3_)	0.88 d (6.6)	1,4	25.5 (CH_3_)	0.88 d (6.6)	1,4
**2**	25.6 (CH_3_)	0.93 d (6.6)	2,4	25.6 (CH_3_)	0.93 d (6.6)	2,4
**3**	25.7 (CH)	1.61 m	2,4,5	25.7 (CH)	1.61 m	2,4,5
**4**	45.3 (CH_2_)	1.90 dd (2.1, 6.1)	3,5,6	45.3 (CH_2_)	1.90 dd (2.1, 6.1)	3,5,6
**5**	85.9 (C)	-	-	85.9 (C)	-	-
**6**	212.5 (C)	-	-	212.5 (C)	-	-
**7**	50.4 (C)	-	-	50.4 (C)	-	-
**8**	214.5 (C)	-	-	214.5 (C)	-	-
**9**	51.2 (C)	-	-	51.2 (C)	-	-
**10**	87.3 (CH)	3.93 s	6,8,9,13,14,1′	86.9 (CH)	3.87 s	6,8,9,13,14,1′
**11**	26.6 (CH_3_)	1.17 s	6,7,8,12	26.6 (CH_3_)	1.17 s	6,7,8,12
**12**	27.4 (CH_3_)	1.22 s	7,10,11	27.4 (CH_3_)	1.22 s	7,10,11
**13**	23.2 (CH_3_)	1.36 s	8,9,10,11	23.2 (CH_3_)	1.36 s	8,9,10,11
**14**	25.2 (CH_3_)	1.23 s	7,8,10,13	25.2 (CH_3_)	1.23 s	7,8,10,13
**1′**	106.2 (CH)	4.48 d (7.3)	10	106.2 (CH)	4.48 d (7.3)	10
**2′**	76.6 (CH)	3.21 t		76.6 (CH)	3.21 t	
**3′**	78.2 (CH)	3.37 ov		78.2 (CH)	3.37 ov	
**4′**	72.6 (CH)	3.30 ov		72.6 (CH)	3.30 ov	
**5′**	75.6 (CH)	3.54 ov		75.6 (CH)	3.54 ov	
**6′**	65.3 (CH_2_)	4.56 dd (2.2, 12.0); 4.35 dd (7.0, 12.0)	9″	65.3 (CH_2_)	4.56 dd (2.2, 12.0); 4.35 dd (7.0, 12.0)	
**1″**	130.8 (C)	-		127.1 ( C )	-	
**2″**	118.0 (CH)	6.80 d (8.4)	3″,4″	116.8 (CH)	6.76 d (8.4)	1″,4″
**3″**	132.0 (CH)	7.44 d (8.4)	1″,2″,4″,7″	134.4 (CH)	7.67 d (8.4)	1″,4″,7″,9″
**4″**	160.8 (C)	-		160.3 (C)	-	
**5″**	132.0 (CH)	7.44 d (8.4)	1″,2″,4″,7″	134.4 (CH)	7.67 d (8.4)	1″,4″,7″, 9″
**6″**	118.0 (CH)	6.80 d (8.4)	3″,4″	116.8 (CH)	6.76 d (8.4)	1″,4″
**7″**	147.6 (CH)	7.63 d (16.1)	1″,8″,9″	145.8 (CH)	6.87 d (12.6)	3″,5″
**8″**	115.7 (CH)	6.32 d (16.1)	1″,7″,9″	116.5 (CH)	5.77 d (12.6)	1″,9″
**9″**	168.9 (C)	-		168.9 (C)	-	

^1^ d = doublet, dd = doublet of doublets; m = multiplet; ov = overlapped; s = singlet; t = triplet *J* values (Hz) are reported in brackets.

## Data Availability

Not applicable.

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
