# Peer review of "Phytochemical Investigation of Myrcianthes cisplatensis: Structural Characterization of New p-Coumaroyl Alkylphloroglucinols and Antimicrobial Evaluation against Staphylococcus aureus"

_plants, 2023, doi:10.3390/plants12051046_

Round 1

Reviewer 1 Report

1、The author mentioned that the antibacterial potential of the methanolic extract of M. cisplatensis leaves against two S. aureus ATCC strains (29213 and 43300) were investigated in Figure 1. Why the inhibitory effects were best at 64 μg/ml rather than at 128 μg/ml. 

2、Single factor analysis of variance should be done for relevant data in Figure 1 and Figure 7.

3. The leaves were collected at flowering stage, what is the growth state of the leaves, juvenile or mature, top or bottom, or the whole plant.

4. Whether the content is different in different growth region and different mature stages?

Author Response

1、The author mentioned that the antibacterial potential of the methanolic extract of M. cisplatensis leaves against two S. aureus ATCC strains (29213 and 43300) were investigated in Figure 1. Why the inhibitory effects were best at 64 μg/ml rather than at 128 μg/ml. 

Ans Considering the standard deviations within the group the results can be considered comparable.

2、Single factor analysis of variance should be done for relevant data in Figure 1 and Figure 7.

Ans Single factor analysis of variance has been performed, the results added in figures 1 and 7. In materials and method section the paragraph -3.6.3 Statistical analysis –has been added.

  1. The leaves were collected at flowering stage, what is the growth state of the leaves, juvenile or mature, top or bottom, or the whole plant.

Ans Most of the leaves were full grown and mature.  The leaves stay on the plant for over a year so it always has mature leaves. The leaves of the whole plant were collected.

  1. Whether the content is different in different growth region and different mature stages?

Ans  In this manuscript we report the phytochemical investigation of leaves collected in Arizona in May 2022. In general,  the cultivated plant in Arizona grows in a climate similar to Uruguay, but drier.  Arizona is hot in Summer (30-40 C common temperatures)  and cool  in winter, with freezing temperatures rare.  M. cisplatensis grows in Arizona well here but needs irrigation.  The plant was growing along a stream in Uruguay. 

Reviewer 2 Report

1. Please provide the name of the botanist who identified the plant material 

2. Most of the methods are without proper citations. Please provide the references

3. "Each tested compound was dissolved in methanol" what is the final concentration of the methanol in the mixture? 

4. Final concentration of the methanol will affect the overall antimicrobial assay? How does the author control this?

5. "MIC was defined as the lowest concentration of drug that caused a total inhibition of microbial growth". How the author determined the MIC concentration. Not clear and please explain this. 

6. If the possible author should conduct statistical analysis for MIC findings.  

7. Please suggest at least one suggestion for future study in the conclusion sections. 

Author Response

  1. Please provide the name of the botanist who identified the plant material 

Ans The plant have been identified by prof. Leslie Landrum, co-author of this paper. Their name has been added in 3.1 section -plant material.

  1. Most of the methods are without proper citations. Please provide the references

Ans Methods should be described or proper citations heve been added

  1. "Each tested compound was dissolved in methanol" what is the final concentration of the methanol in the mixture? 

Ans Each tested compound was dissolved in methanol (1mg/200µl) to give a stock solution and diluted in Mueller-Hinton (MH).

  1. Final concentration of the methanol will affect the overall antimicrobial assay? How does the author control this?

Ans Each experiment included negative control wells containing bacteria in Mueller–Hinton broth plus the amount of vehicle (methanol) used to dilute each compound. We added a sentence in the manuscript, results section, paragraph 2.1, lines 86-87

  1. "MIC was defined as the lowest concentration of drug that caused a total inhibition of microbial growth". How the author determined the MIC concentration. Not clear and please explain this. 

Ans The compounds were added to the bacterial suspension in each well yielding a final cell concentration of 1 × 106 CFU/ml and a final compound concentration ranging from 128 to 16µg/ml. The procedure was reported as previously published. Anyway, we added a sentence in material and methods section to better specify the assay. Paragraph 3.6.2

  1. If the possible author should conduct statistical analysis for MIC findings. 

Ans Statistical analysis has been added in the figures 1 and 7, and in test. Furtheremore in materials and method section the paragraph -3.6.3 Statistical analysis –has been added.

  1. Please suggest at least one suggestion for future study in the conclusion sections. 

Ans In conclusion section, we have included one  suggestion for future plan.

Reviewer 3 Report

The manuscript is interesting and well drafted, however, there are some suggestions that need to be incorporated before publication.

Author Response

  1. Family name should be in italic font

Ans. Done. Myrtaceae has been italicize in manuscript

  1. Line 20 also in Brazil and Paraguay as per the following link

https://powo.science.kew.org/taxon/urn:lsid:ipni.org:names:30076995-2

Ans. Thanks for your suggestion. The sentence about plant distribution has been added in abstract as well as in introduction section.

  1. Line 27 How the authors have analyzed this MIC value?

Ans The procedure was reported as previously published. Anyway, we added a sentence in material and methods section to better specify the assay. Paragraph 3.6.2

  1. Line 40

Ans Correction has been made

  1. Lines 85-86Rewrite . complex statement lines 84-85

Ans The sentence has been rephrased

  1. Line 87 what about another control oxacillin

Ans Oxacillin was not able to inhibit the growth of the MRSA (data not showed). we added this sentence in paragraph 2.1

  1. Line 245 Add Statistical analysis part in methodology

Ans Statistical analysis has been added in the figures 1 and 7, and in test. Furtheremore in materials and method section the paragraph -3.6.3 Statistical analysis –has been added.

  1. Line 274 Have u measured the particle size?

Ans For our extraction method, a precise measure of particle size in not necessary. The pulverization is functional to increase the exchange surface between the plant material and the solvent

  1. Lines 368-368 Elaborate the methodology

Ans The procedure was reported as previously published- ref- 14. Anyway, we added a sentence in material and methods section to better specify the assay. Paragraph 3.6.2

10-. Line 375 Draw significant conclusion of your research, not generalized statements here

 Ans Thanks for your suggestion. We insert a specific conclusion for this manuscript

  • Line 379 Repetition of the intro line

Ans The sentence has been deleted

Round 2

Reviewer 3 Report

All the suggestions have been incorporated.